# Li/Na Ion Storage Performance of a FeOF Nano Rod with Controllable Morphology

Linhua Li [1], Liangshun Xiang [2], Yan Lin [1,*], Lei Chen [1], Renqing Guo [1], Yiqi Cao [1], Xiaohua Huang [1] and Jianbo Wu [1]

1   Zhejiang Provincial Key Laboratory for Cutting Tools, Taizhou University, Taizhou 318000, China; lilinhua0816@163.com (L.L.); Leichenah@163.com (L.C.); guorenq@tzc.edu.cn (R.G.); caoyqtzc@163.com (Y.C.); huangxn@tzc.edu.cn (X.H.); wujb@tzc.edu.cn (J.W.)
2   Zhejiang Nandu Power Supply Co., Ltd., Hangzhou 310012, China; xiangliangshun@naradapower.com
*   Correspondence: linyan@tzc.edu.cn

**Abstract:** Although the conversion material iron oxyfluoride (FeOF) possesses a high theoretical specific capacity as a cathode material for Li/Na ion batteries, its poor rate and cycling performances, caused mainly by sluggish ($Li^+$/$Na^+$) reaction kinetics, restrict its practical application. Herein, FeOF with high purity, a fusiform nanorod shape and high crystallinity is prepared through a facile chemical solution reaction. The electrochemical measurements show that the present FeOF exhibits high capacity and good cycling stability as a cathode material for Li-ion batteries. Capacities of 301, 274, 249, 222, and 194 mAh/g at stepwise current densities of 20, 50, 100, 200, and 400 mA/g are achieved, respectively. Additionally, the capacity at 100 mA/g retains 123 mAh/g after 140 cycles. Meanwhile, as a cathode material for Na ion battery, it delivers discharge capacities of 185, 167, 151, 134 and 115 mAh/g at stepwise current densities of 20, 50, 100, 200, and 400 mA/g, respectively. A discharge capacity of 83 mAh/g at 100 mA/g is achieved after 140 cycles. The excellent lithium/sodium-storage performance of the present FeOF material is ascribed to its unique nanostructure.

**Keywords:** iron oxyfluoride (FeOF); li-ion batteries; sodium ion batteries; electrochemical performance

## 1. Introduction

Nowdays, it is urgent to develop highly efficient energy storage systems to meet the increasing demand for sustainable energy storage. Among them, lithium-ion batteries (LIBs) and sodium-ion batteries (SIBs), which are especially attractive in the electric vehicle market and grid-scale energy storage, are receiving intensive attention due to their high energy/power density [1–3]. Furthermore, the next generation of energy storage systems imperatively requires the development of batteries with excellent performances, e.g., high energy, high power, enhanced safety and improved cyclability. It is known that the electrode material is one of the most critical components determining the performance of Li/Na-ion batteries. Therefore, it is crucially important to develop new electrode materials with excellent performance.

Cathode materials with high capacity and low cost are vital to improving the energy density of commercial batteries for electric devices [1,2]. Compared with traditional intercalation cathodes, such as $LiCoO_2$, the fluoride [4–8] and sulfide [9–12] conversion cathodes have a much higher capacity due to their accommodation of more than one $Li^+$ per transition metal core during the charge/discharge process, which makes them potential cathode materials for LIBs. Among them, the mixed-anion cathode material FeOF was especially studied in Li-ion batteries [13–16] and Na-ion batteries [17–19]. FeOF has the combined advantages of fluorides, with an impressively high theoretical specific capacity of 885 mAh/g and charge/discharge voltages, and the superiority of oxides. However, its disadvantages of poor rate capability and low Coulombic efficiency caused by low electronic conductivity and ion diffusion capability urgently need to be surmounted.



Inspired by Deng [20], we propose a facile preparation method for high-purity FeOF with a nanorod shape by a mild solvothermal reaction with commercial $FeF_3 \cdot 3H_2O$ and 1-propanol as the only reactant and solvent. Although FeOF has been investigated as a cathode candidate for LIBs, its electrochemical performance for applications in SIBs lacks detailed studies. Herein, we report the electrochemical performance of the pure FeOF nanorod as Li/Na-storage cathode material in this paper. Iron fluorides are also considered promising candidates for LIBs and SIBs [5,13]. Therefore, the electrochemical performance of $FeF_3 \cdot 3H_2O$ used in the preparation of FeOF is also studied for comparison. Results show that the nanorods are composed of single-phase FeOF with high crystallinity and exhibit excellent Li/Na-storage performance, while the $FeF_3 \cdot 3H_2O$ demonstrates poor but reversible Li/Na-storage performance.

## 2. Materials and Methods

### 2.1. Preparation of FeOF Nanorods

FeOF nanorods were prepared by a modified solvothermal method according to the literature [20,21]. In a typical synthesis, 147 mg of $FeF_3 \cdot 3H_2O$ was ground in 1-proponal solution with a mortar and pestle for 30 min to reduce the particle size. Then, extra 1-proponal was added to cause the solution to reach 75 mL. The solution was stirred vigorously at room temperature for half an hour and subsequently transferred into a 100 mL stainless-steel autoclave, which was then Teflon-sealed and heated in an oven at 175 °C for 20 h. Pure FeOF was then generated after being centrifuged, washed with ethanol three times and dried in air at 75 °C overnight. Analytical-grade chemicals were used as obtained in all experiments.

### 2.2. Material Characterizations

The phase identification and the crystallinity of the prepared products was determined by X-ray diffraction (XRD, Bruker D8, Taizhou, China) with a Cu Ka radiation at a scan rate of 2°/min in the 2θ range of 10–90°. The morphology and particle size of the FeOF sample were observed by a field emission scanning electron microscope (SEM, S-4800, Taizhou, China).

### 2.3. Electrochemical Measurements

The electrode was prepared by spreading the homogeneous N-methylmethyl-pyrrolidone slurry containing 70 wt% as-synthesized FeOF, 15 wt% acetylene black and 15 wt% polyvinylidene fluoride (PVDF) onto an aluminum foil current collector, and then dried at 80 °C for 12 h. Generally, the active materials loaded on the current collector were around 0.95 mg/cm$^2$. Coin cells of the CR2025 type were assembled in an argon-filled glove box. For LIBs, metallic lithium was used as the anode, 1 M $LiPF_6$ in dimethyl carbonate/ethylene carbonate (1:1 vol%) as the electrolyte and Celgard 2300 film as the separator. For SIBs, metallic sodium was used as the anode, 1 M $NaClO_4$ in propylene carbonate/ethylene carbonate/diethyl carbonate (1:1:1 vol%) as the electrolyte and Whatman glass microfiber filters as the separator. The amount of the electrolytes used both in LIBs and SIBs was about 0.5 mL. The charge/discharge properties of all electrodes were tested at different rates between 1.2 and 4.0 V on the Land test instrument. The rate performance was performed at current densities of 20, 50, 100, 200, and 400 mA/g, respectively. Then, the cycling performance test was performed at 100 mA/g for 140 cycles. The cyclic voltammetry (CV) and the electrochemical impedance spectroscopy (EIS) were executed by a CHI660E Electrochemical Workstation. The CV curves were measured between 1.2 and 4.0 V at a scanning rate of 0.1 mV/s, and the amplitude of EIS was 5 mV within the frequency range of 0.1–100 KHz. Room temperature of 25 ± 5 °C was applied in all the electrochemical measurements.

## 3. Results

The overall synthetic process of an FeOF nanorod is shown in Figure 1. By solvothermal treatment, oxygen atoms in 1-propanol replace the fluorine atoms in FeF$_3$. Reactions involved in the evolution of pure FeOF could possibly be [20]:

$$FeF_3 \cdot 3H_2O \rightarrow FeF_3 + 3H_2O \tag{1}$$

$$FeF_3 + xHOR \rightarrow FeF_{3-x}(OR)_x + xHF \tag{2}$$

$$FeF_{3-x}(OR)_x + (2 - x)H_2O \rightarrow FeF(OR)_x(OH)_{2-x} + (2 - x)HF \tag{3}$$

$$FeF(OR)_x(OH)_{2-x} \rightarrow FeOF + xHOR + (1 - x)H_2O \tag{4}$$

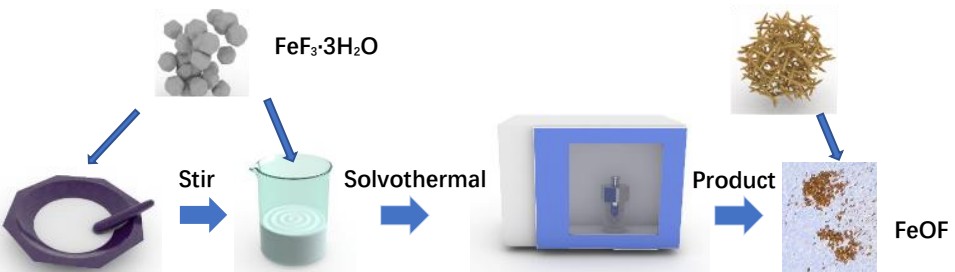

**Figure 1.** Schematic process of the formation of FeOF nanorods.

The XRD patterns of the precursor FeF$_3$·3H$_2$O and the synthesized FeOF are shown in Figure 2. All the observed reflections of the synthesized sample can be indexed to that of FeOF (JCDS card No. 70-1522) with a tetragonal rutile structure. The precursor FeF$_3$·3H$_2$O is also composed of the pure FeF$_3$·3H$_2$O phase. Both the precursor and the obtained sample are crystallized well, as seen from the XRD results. The crystal size of both the reactant and the product are listed in Table 1. For each material, the crystal size changes slightly with the facet index. However, the grain size of FeF$_3$·3H$_2$O material is obviously larger.

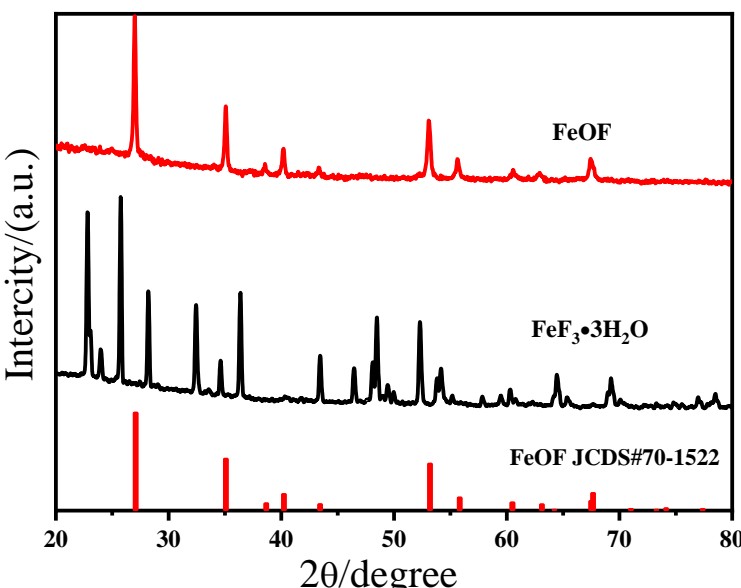

**Figure 2.** XRD patterns of the precursor and the synthesized sample.

**Table 1.** Structural parameters of FeOF and FeF3·3H2O determined from XRD data.

| Sample | *[hkl]* Values | 2θ (°) | FWHM (°) | Grain Size (Å) |
|---|---|---|---|---|
| FeOF | 110 | 26.9 | 0.377 | 226 |
| | 101 | 35.2 | 0.268 | 340 |
| | 211 | 53.1 | 0.396 | 233 |
| FeF$_3$·3H$_2$O | 110 | 16.1 | 0.200 | 473 |
| | 200 | 22.8 | 0.165 | 638 |
| | 101 | 25.8 | 0.173 | 594 |

The morphologies of the sample and the FeF$_3$·3H$_2$O precursor are shown in Figure 3. The SEM image (Figure 3b) shows that the morphology of bare FeOF is a nanorods with a fusiform shape, which is the typical morphology of FeOF synthesized by the solvothermal method [20]. The particle size of the fusiform nanorod varies from 50 to 75 nm in width and from 213 to 325 nm in length. Combined with the XRD results, the small particles present in Figure 3b are probably the ungrown-up or broken FeOF particles. The particles of the precursor FeF$_3$·3H$_2$O aggregate severely and have a much larger particle size, typically 500 nm in width to 1.5 μm in length.

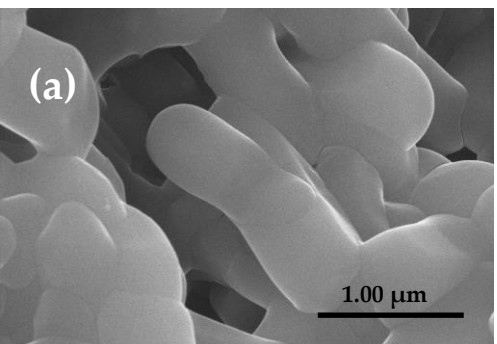
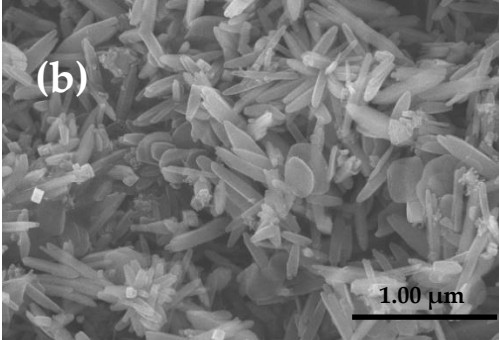

**Figure 3.** SEM graphs of FeF$_3$·3H$_2$O (**a**) and FeOF (**b**).

### 3.1. Electrochemical Performances in Li-Ion Battery

Figure 4a displays the cyclic performance of the sample at various current densities of 20, 50, 100, 200, and 400 mA/g, respectively, for the first 11 cycles and following 140 cycles at 100 mA/g within the voltage range of 1.2–4.0 V. The initial discharge and charge capacities of the FeOF electrode are 450 mAh/g and 390 mAh/g, respectively, demonstrating a relatively high Coulombic efficiency of 87% in conversion cathode systems. The Coulombic efficiency remains almost 99% on average during the cycling at a density of 100 mA/g. However, the specific capacity in the following 5 cycles shows a sharp decay, which may be due to the irreversible side reactions and the irreversible formation of a cathode electrolyte interphase (CEI) on the surface of the electrode [22]. This can be further proved by the EIS results in Figure 4e,f. After a careful evaluation, different equivalent circuits are applied for each curve. For the electrode before cycling, the equivalent circuit is composed of R$_e$, which is attributed to the Ohmic resistance of the electrolyte, a Zarc element (CPE and R$_{ct}$, attributing to the double-layer capacitance and charge transfer resistance) and a Warburg impedance W$_1$. Besides the elements mentioned above, the equivalent circuit for the electrode after cycling composes one more Zarc element (CPE and R$_{CEI}$), which is attributed to CEI capacitance and resistance. The R$_{CEI}$ and R$_{ct}$ values determined by EIS spectra are 0, 83.3 Ohm and 39.1, 85.5 Ohm for the FeOF electrode before and after cycling, respectively. The increase of R$_{CEI}$ demonstrates that CEI probably forms during the charge/discharge process, which is in accordance with the cycling results.

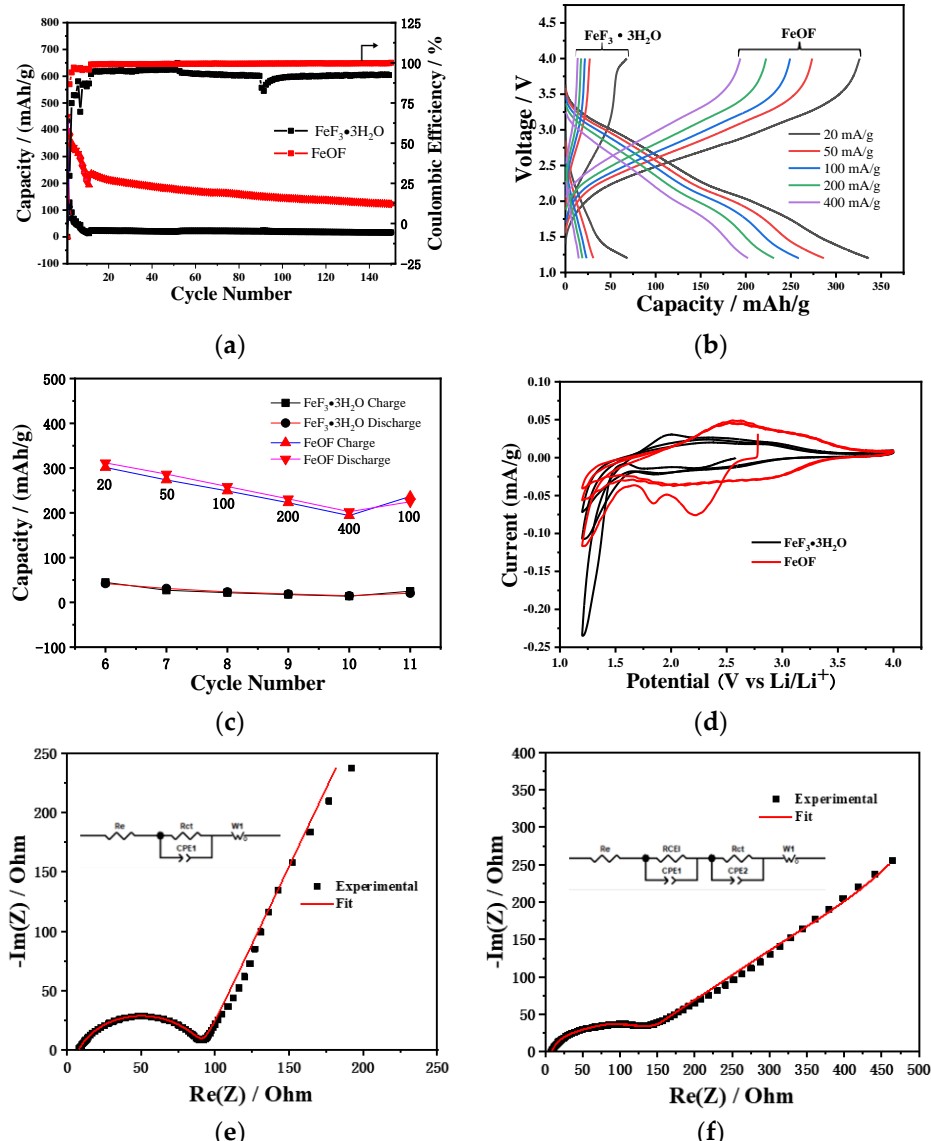

**Figure 4.** (**a**) Coulombic efficiency and the cycling performance. (**b**) The charge/discharge profiles at different current rates. (**c**) Charge/discharge capacities at different current rates. (**d**) CV curves at 1.2–4.0 V. (**e**) EIS curves of the FeOF electrodes for LIBs before cycling. (**f**) EIS curves of the FeOF electrodes for LIBs after cycling. The equivalent circuit used for fitting is shown in the inset.

Figure 4c shows the charge/discharge capacities at different current densities of FeOF. As can be seen, the charge capacities of FeOF at stepwise current densities of 20, 50, 100, 200, and 400 mA/g are 301, 274, 249, 222 and 194 mAh/g, respectively. While the current densities return to 100 mA/g after 400 mA/g, the charge capacity increases to 236 mAh/g, corresponding to a capacity retention of 95%. After 140 cycles, the charge capacity at 100 mA/g decays to 123 mAh/g with a capacity retention of 52%.

Table 2 compares the cycling performance of the present FeOF with related representative FeOF cathode materials for LIBs reported in the literature. It is seen that the present FeOF has an overall good electrochemical performance, which is mainly ascribed to its nanorod morphology and high FeOF purity.

**Table 2.** Comparisons of the electrochemical performances of the present FeOF with related representative FeOF cathode materials for LIBs reported in the literature.

| Sample Number | Voltage (V vs. Li/Li+) | Current Density (mA/g) | Capacity (mAh/g) | Cycle Number | Reference |
|---|---|---|---|---|---|
| 1 | 1.2–4.0 | 100 | 178 | 50 | This work |
|   |         | 100 | 145 | 100 |           |
| 2 | 1.0–4.0 | 100 | 104 | 50 | [15] |
| 3 | 1.2–4.0 | 300 | 75 | 100 | [23] |
| 4 | 2.0–4.0 | 10 | 180 | 30 | [24] |
| 5 | 2.0–3.8 | 100 | 45 | 100 | [25] |

The charge/discharge profiles of the first 10 cycles at different current densities of 20, 50, 100, 200 and 400 mA/g, respectively, are illuminated in Figure 4b to further analyze the electrochemical properties. It can be seen obviously that two plateaus are present in all discharge curves. The plateau at approximately 2.75 V is attributed to the intercalation of Li into FeOF crystal structure, and the other one at about 2.0 V is attributed to the conversion reaction from LiFeOF to LiF, $Li_2O$ and Fe. Conversely, two plateaus at ca. 2.5 and 3.0 V are also observed in all charge curves, corresponding to the occurrence of the reaction of LiFeOF to FeOF as well as the inverse transform from LiF, $Li_2O$ and Fe to LiFeOF, respectively. Combining the results of Figure 4b,c, the specific energy can be obtained to be ca. 622 mWh/g at 100 mA/g, which is a little smaller than that of LFNCNP (656 mWh/g) reported by Pagot [26], but larger than that of the material reported in another paper [27].

The cyclic voltammetry (CV) curves in the voltage range of 1.2–4.0 V (vs. Li/Li+) for the first three cycles are tested and shown in Figure 4d to understand the lithium storage mechanism of the FeOF electrode. It can be distinctly observed that the FeOF electrode exhibits two reduction peaks in the cathodic scanning and two oxidation peaks in the anodic scanning, which is in accordance with the above-mentioned charge/discharge plateaus in Figure 4b. Except for the 1st cycle in the CV curve, a large reduction peak is seen at around 2.9 V, which is ascribed to the intercalation of Li+ into the FeOF crystal structures, corresponding to the plateau at 2.75 V in the discharge curve (Figure 4b). The reduction peak centered at ca. 2.0 V is attributed to the conversion reaction from LiFeOF to LiF, Fe and $Li_2O$, corresponding to the plateau at 2.0 V in the discharge curves (Figure 4b). Besides, in the anodic scanning, two oxidation peaks at about 2.7 and 3.1 V are also obviously shown. These two peaks correspond to the oxidation of LiFeOF to FeOF and the oxidation of $Li_2O$, LiF and Fe to LiFeOF, respectively. It can be concluded from the above results that the improved electrochemical properties can mainly be ascribed to the nanorod structure of the FeOF material, which can effectively shorten the diffusion distance of Li+ and redeem the volume variation caused by the repeated Li+ insertion/extraction.

The tap density of this nanorod FeOF is ca. 2.1 g/cm³, which is much lower than that of $LiCoO_2$ [28]. Although the tap density of nanostructured materials is a major disadvantage for practical applications, which is not conducive to the improvement of volume energy density, this nanorod structure is still attractive. Therefore, to overcome the shortcoming of low tap density of the FeOF nanorod, multiscale particles with built-in FeOF nanorod are a promising way to improve the tap density and electrochemical performance at the same time in future work [29].

For comparison, the electrochemical properties of the $FeF_3 \cdot 3H_2O$ electrode are also shown in Figure 4. It is apparently seen that the $FeF_3 \cdot 3H_2O$ electrode demonstrates much poorer electrochemical performance, although it can lithiate and delithiate reversibly, demonstrating a capacity of 50 mAh/g at 20 mA/g.

### 3.2. Electrochemical Performances of FeOF in Na-Ion Battery

The Na-ion storage performance of FeOF was also conducted with the same galvanostatic charge/discharge measurement in the LIBs. As displayed in Figure 5a, the charge

capacity at 20 mA/g of the first cycle is 204 mAh/g, which decays to 185 mAh/g in the 6th cycle. The rate capability of FeOF as the cathode for SIBs is further characterized in Figure 5c. It is seen that the charge capacities are 185, 167, 151, 134 and 115 mAh/g at stepwise current densities of 20, 50, 100, 200, and 400 mA/g, respectively. While the current rate returns to 100 mA/g after 400 mA/g, the charge capacity shows almost no decay. However, it decays to 83 mAh/g after 140 cycles during the cycling performance at 100 mA/g, demonstrating a capacity retention of 55%. The Coulombic efficiency remains above 99% on average (Figure 5a). Table 3 compares the electrochemical properties between the nanorod FeOF and other reported bare FeOF as cathode materials for SIBs. The nanorod FeOF exhibits preferable electrochemical performance compared with other reported bare FeOF, demonstrating the largest capacity of 100 mAh/g at 100 mA/g after 100 cycles. The nanorod morphology and high FeOF purity of the material in this work have profound benefits in improving the electrochemical performance.

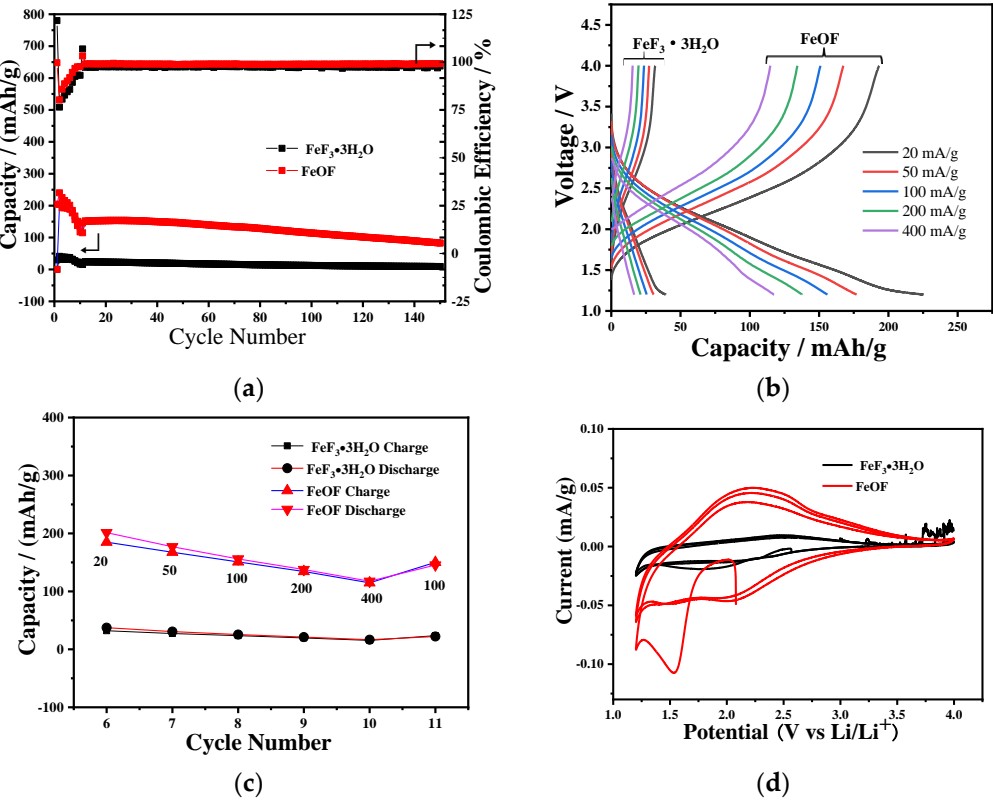

**Figure 5.** (**a**) Coulombic efficiency and cycling performance. (**b**) The charge/discharge profiles at different current rates. (**c**) Charge/discharge capacities at different current rates. (**d**) CV curves at 1.2–4.0 V.

**Table 3.** Comparisons of the electrochemical performances for different bare FeOF as cathode material for SIBs.

| Sample Number | Voltage (V vs. Li/Li⁺) | Current Density(mA/g) | Capacity (mAh/g) | Cycle Number | Reference |
|---|---|---|---|---|---|
| 1 | 1.2–4.0 | 20 | 201 | 6 | This work |
|   |         | 100 | 100 | 100 |  |
| 2 | 1.2–4.0 | 100 | 20 | 100 | [18] |
| 3 | 1.0–4.0 | 20 | 210 | 20 | [20] |
| 4 | 1.2–4.0 | 20 | 20.4 | 50 | [30] |
| 5 | 1.2–4.0 | 100 | 53.2 | 100 | [17] |

Figure 5b shows the charge/discharge profiles of FeOF and $FeF_3 \cdot 3H_2O$ at different current densities of 20, 50, 100, 200 and 400 mA/g, respectively. Two plateaus can be seen from all the discharge curves, which are due to the intercalation of Na into the FeOF crystal structure at ca. 2.0 V, and the conversion reaction from NaFeOF to NaF, $Na_2O$ and Fe at about 1.5 V. Apparently, the output voltage of the FeOF material is much lower than that in LIBs. The charge curves of SIBs only show a less obvious plateau at ca. 3.0 V due to the conversion reaction from NaF, $Na_2O$ and Fe to NaFeOF within the same voltage range from 1.2–4.0 V, which is an obvious difference compared to LIBs. The less obvious conversion reaction during the charging process is probably related to the fact that lithium ions have a much smaller radius than sodium ions. Thus, an intercalation reaction is preferred in SIBs to avoid large volume variation, thereby maintaining a stable structure during the charge/discharge process to obtain good cycling performance [13]. Combining the results of Figure 5b,c, the specific energy obtained is ca. 272 mWh/g at 100 mA/g, which is much smaller than LIBs. However, it is still attractive in SIBs, as seen in Table 3.

The CV tests are carried out to explore the mechanisms of Na-ion storage. Figure 5c shows the CV results of FeOF and $FeF_3 \cdot 3H_2O$ electrodes for the initial three cycles at a scan rate of 0.1 mV/s between 1.2 and 4.0 V (vs. $Na/Na^+$). As can be seen, $FeF_3 \cdot 3H_2O$ electrodes have one reduction and oxidation peak found at 1.7 V and 2.2 V, respectively, showing that the sodiation process is reversible. During the first cathodic scanning, two reduction peaks are seen at about 2.1 V and 1.5 V for the FeOF electrode. These peaks are ascribed to the intercalation reaction of FeOF to NaFeOF and the conversion reaction of NaFeOF to NaF, $Na_2O$ and Fe, respectively. In the following cycles, these two peaks become much smaller, implying the capacity loss in the first cycle. Part of the capacity is consumed in the generation of CEI films on the cathode surface and other irreversible side reactions. In the anodic scanning, an oxidation peak is found at ca. 2.2 V, which is attributed to the deintercalation reaction of NaFeOF to FeOF. However, the oxidation peak centered at ~2.7 V corresponding to the conversion reaction becomes negligible, indicating that the conversion reaction contributes much less capacity. This is well consistent with the charge/discharge results of FeOF shown in Figure 5b. In addition, the redox peaks in the following cycles mostly overlap, indicating that the nanorod FeOF possesses favorable cyclic stability.

## 4. Conclusions

FeOF, with its fusiform nanorod shape and high crystallinity, was synthesized from a $FeF_3 \cdot 3H_2O$ precursor through a one-step solvothermal method with no subsequent sintering. The FeOF material endows favorable electrochemical properties as cathode materials for lithium- and sodium-ion batteries. It can deliver a high initial discharge capacity of 249 and 194 mAh/g at current densities of 100 and 400 mA/g, respectively, and retain 123 mAh/g after 140 cycles at 100 mA/g at 1.2–4.0 V vs. $Li/Li^+$. Its specific energy is ca. 622 mWh/g at 100 mA/g. As a cathode material of SIBs, it possesses high capacities of 151 and 115 mAh/g at 100 and 400 mA/g, respectively, and a specific energy of ca. 272 mWh/g at 100 mA/g as well. Meanwhile, it demonstrates good cycling stability, with a discharge capacity of 83 mAh/g at 100 mA/g still achieved after 140 cycles. The favorable Li/Na-storage properties should be ascribed to the unique structure of FeOF, which is beneficial to shortening the diffusion pathway of $Li^+/Na^+$ and enhancing the structural stability of the material.

**Author Contributions:** Conceptualization, Y.L., L.X. and J.W.; methodology, Y.L., X.H. and L.C.; validation, L.X. and Y.C.; investigation, L.L., Y.L. and L.X.; resources, R.G.; data curation, Y.L., L.C. and R.G.; writing—original draft preparation, L.L., Y.L., L.X. and L.C.; writing—review and editing, Y.L., R.G. and X.H.; supervision, J.W.; project administration, L.X. and R.G.; funding acquisition, Y.C. and X.H. All authors have read and agreed to the published version of the manuscript.

**Funding:** This work was financially supported by the Zhejiang Province Public Welfare Technology Application Research Project, PR China, under Grant No. LGG22E010003 and the Natural Science Foundation of Zhejiang Province, PR China, under Grant No. LTY20E010001.

**Institutional Review Board Statement:** Not applicable.

**Informed Consent Statement:** Not applicable.

**Data Availability Statement:** Not applicable.

**Conflicts of Interest:** The authors declare no conflict of interest.

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
