# Peer review of "Li/Na Ion Storage Performance of a FeOF Nano Rod with Controllable Morphology"

_processes, doi:10.3390/pr10081491_

Round 1

Reviewer 1 Report

Nanostructured materials are not feasible for practical applications because of this low tap density, as highlighted in recent papers (Nat Rev Mater (2022). https://doi.org/10.1038/s41578-022-00454-9); The author should comment on how to address the low tap density of your nanostructured FeOF from the practical application viewpoint.

What is the real density of this material?

What is the highest tap density of FeOF electrodes?

Author Response

The reviewer thought that nanostructured materials are not feasible for practical applications and the author should comment on how to address the low tap density of the nanostructured FeOF from the practical application viewpoint.

The real density of nanocrystalline materials is largely unknown for most nanostructured materials are in a form of powder which makes the determination of their density difficult [1]. The theoretical density of FeOF material is 4.53 g/cm3 from XRD data, which must be far larger than its real density. It’s tap density in our work is about 2.1 g/cm3.

The tap density of nanostructured materials is indeed a major disadvantage for practical applications, which is not conducive to the improvement of energy density, just as the reviewer thought. Nevertheless, in the reference suggested by the reviewer, it is pointed out that multiscale particles where microscale particles with built-in nanoscale features, such as microparticles assembled from nanoscale building blocks or patterned with engineered or natural nanopores, offer exciting possibilities to develop battery electrodes [2]. This also indicates the importance of building nanostructured materials. Therefore, to overcome the shortcoming of low density of FeOF nanorod, we are to design a multiscale particles with built-in FeOF nanorod which can improve the tap density and electrochemical performance at the same time in the next work.

[1] T. D. Shen, Jianzhong Zhang, Yusheng Zhao. What is the theoretical density of a nanocrystalline material? Acta Materialia, 56 (2008) 3663-3671.

[2] R. Jain, A.S. Lakhnot, K. Bhimani, S. Sharma, V. Mahajani, R.A. Panchal, M. Kamble, F. Han, C. Wang, and N. Koratkar, Nanostructuring versus microstructuring in battery electrodes. Nature Reviews Materials, 2022.

Reviewer 2 Report

This work is about a novel cathode material for both lithium and sodium battery applications. The topic is interesting, and the English language is of a sufficient level for publication. The results are promising. The experimental part is lacking of some experiments. Overall, I suggest to accept this Manuscript only after a Major Revision.

Here below are summarized the points that must be discussed during the revision:

1.      XRD results: can the authors determine the crystal size of both the reactant and the product (Scherrer equation)? Are there any differences? Are there any differences if different reflections are used in the Scherrer equation?

2.      Lines 125-127: this hypothesis must be confirmed with EIS measurements before and after cycling the battery device.

3.      CV results: CV curves and charge/discharge profiles are not perfectly in agreement, in terms of peak area (CV) and plateau length (C/D profiles). This may be due to the cell setup used in the CV experiments: a three-electrodes cell must be used.

4.      Conclusions: can the authors compare the specific energy of this cathode vs that of conventional and high-voltage cathodes (see for example DOI: 10.1016/j.electacta.2016.12.149; DOI: 10.1038/s41560-018-0107-2; DOI: 10.1039/d0ta06865a; DOI: 10.1002/adfm.200600442)? Results are interesting in terms of specific capacity, but also the specific energy value must be considered.

Author Response

  1. According to the reviewer’s suggestion, the crystal size of both the reactant and the product obtained from XRD results are listed in Table 1. For each material, the crystal size changes slightly with facet index. However, the grain size of FeF33H2O material is obviously larger.

Table 1 Structural parameters of FeOF and F FeF3•3H2O determined from XRD data

Sample

[hkl] values

2θ(º)

FWHM(º)

Grain Size(Å)

FeOF

110

26.9

0.377

226

101

35.2

0.268

340

211

53.1

0.396

233

FeF3•3H2O

110

16.1

0.200

473

200

22.8

0.165

638

101

25.8

0.173

594

  1. The reviewer pointed out that Lines 125-127: this hypothesis must be confirmed with EIS measurements before and after cycling the battery device. Figure 1 shows EIS curves of the FeOF electrodes for LIBs before and after cycling. The equivalent circuit used for fitting is shown in the inset. After a careful evaluation, different equivalent circuits are applied for each curve. For the electrode before cycling, the equivalent circuit is composed of Re which is attributed to the Ohmic resistance of the electrolyte, a Zarc element (CPE and Rct attributing to the double-layer capacitance and charge transfer resistance) and a Warburg impedance W1. Besides the elements mentioned above, the equivalent circuit for the electrode after cycling composes one more Zarc element (CPE and RCEI which is attributed to CEI capacitance and resistance, respectively). RCEI, Rct values determined by EIS spectra are 0, 83.3 Ohm and 39.1, 85.5 Ohm, respectively, for the FeOF electrode before and after cycling. The increase of RCEI demonstrates that CEI probably forms during the initial charge/discharge process, which may cause a sharp capacity decay. The EIS results are in accordance with the cycling results.

(a)

 (b)

Figure1. (a) EIS curves of the FeOF electrodes for LIBs before cycling. (b) EIS curves of the FeOF electrodes for LIBs after cycling. The equivalent circuit used for fitting is shown in the inset.

  1. The reviewer thought that CV curves and charge/discharge profiles are not perfectly in agreement in terms of peak area (CV) and plateau length (C/D profiles), and suggested a three-electrodes cell must be used in the CV experiments.

In this paper, CV is used to obtain the approximate electrode reaction mechanism, that is, whether the number of redox peaks in CV is consistent with the number of charge/discharge platforms, and the potentials of lithiation/delithiation processes corresponding to the different mechanisms. The data obtained from the existing two-electrodes is enough to explain the academic problem to be solved in this paper. Moreover, a large part of the existing academic research uses two-electrodes in CV test. Of course, no matter how complex or precise the measurement method is, it is not too much for academic research.

However, the reviewer also pointed out new research methods. In our further study of the application of a new material in secondary batteries, we can use three electrodes to study the CV performance in detail.

  1. According to the reviewer’s suggestion, the specific energy of FeOF material as LIBs in this work calculated is ca. 622 mWh/g at 100 mA/g, which is a little smaller than that of LFNCNP (656 mWh/g) reported by Pagot [1], but larger than that of the material reported in another papers [2]. Meanwhile, the specific energy of FeOF material as SIBs is ca. 272 mWh/g which is much smaller than that as LIBs, it is still attractive in SIBs seen from the Table 3 in the article.

[1]. Pagot, G., et al., High valence transition metal-doped olivine cathodes for superior energy and fast cycling lithium batteries. Journal of Materials Chemistry A, 2020. 8(48): p. 25727–25738.

[2]. Pagot, G., et al., Effect of Graphite and Copper Oxide on the Performance of High Potential Li Fe1/3Ni1/3Co1/3 PO4 Olivine Cathodes for Lithium Batteries. Electrochimica Acta, 2017. 225: p. 533-542

Round 2

Reviewer 2 Report

The revised manuscript is acceptable for publication.

This manuscript is a resubmission of an earlier submission. The following is a list of the peer review reports and author responses from that submission.

Round 1

Reviewer 1 Report

Although the authors could successfully synthesize FeOF material utilizing as the electrode material of LIB and SIB, a strong discussion has not been provided regarding the performance of the batteries. In addition, the context is suffering from incoherency, and some grammatical mistakes and typos as well. I am afraid if these flaws can be corrected in the time frame of a major revision. Please find the comments below:

1. The numerical data in the abstract is a bit confusing and not comparable. For Li-ion battery, the initial capacity and capacity after 140 cycles at a current density of 100 mA/g are reported; while for Na-ion battery it is in a different way. It might be better if you can report some comparable dataonf two sorts of batteries.

2. Please check the manuscript carefully and modify the grammatical mistakes and typos. For instance, the sentence of “Compared with traditional intercalation cathode, such as LiCoO2, the fluoride [3-7] and sulfide [8-11] conversion cathodes have much higher capacity for they can accommodate more than one Li+ per transition metal core during charge/discharge, which makes them become potential cathode material for LIBs” lacks fluency. Some other examples: a impressively (line 44), and charge/discharge voltages, and the superiority of oxides (line 45), we propose ta facile preparation (line 48), for LIBs and SIBs are also studied, for iron fluorides are also considered as promising candidates (line 55), etc.

3. The introduction section can be improved by reporting some numerical data of conventional electrodes and also listing recently used novel electrodes. These recent publications might be helpful: https://doi.org/10.1002/sstr.202100217 and https://doi.org/10.3390/en15031162

4. Could you please explain why did you use such a low scan rate (0.1 mV/s) for CV test?

5. Having a discussion about the involved reactions during the synthesize is recommended.

6. Figure 4 and 5 are in an inappropriate situation and should be rearranged to occupy less space.

7. After talking about Figure 4a, you missed Figure 4b.

8. The first two sentences in section 3.2. can be eliminated since they are suitable for the introduction section, not discussion.

9. Considering CV of the electrodes for SIB, is observing two weak oxidation and reduction peaks at 1.7 and 2.2 V, respectively, enough to claim the process is reversible?

10. Why there is no discussion about the EIS tests?

Reviewer 2 Report

Please see my comments in the attachment

Reviewer 3 Report

The paper entitled “Li/Na ion storage performance of FeOF nano rod with controllable morphology” by Linhua Li et.al. clearly explains the good electrochemical performance of the FeOF nanorod structure. The overall structure of the study is good. After the below mentioned corrections, I recommend accepting the publication.

1.      In line 30, the authors mention the use of Sodium-ion batteries in future electric vehicle applications, however, the references do not say much about it, may be the authors can provide references where Sodium-ion batteries applied to EVs or maybe reframed it as a good candidate for grid application.

2.      It is mentioned that FeOF has a high theoretical specific capacity of 885 mAh/g, however, did not mention it w.r.t Li/Li+ or Na/Na+. I highly recommend doing it.

3.      It is important to mention the values of the purity of the chemicals used in the techniques since they affect the phase purity at the end.

4.      Authors should explain the reason for changing the separators for Lithium and sodium-ion batteries (in terms of wettability) and also the amount of electrolyte used.

5.      Authors should explain the reason for doing only 1 cycle at each current rate at the C-rate test after the formation cycle.? What is the capacity difference between the 1st and 2nd cycles in the C-rate test?

6.      Authors should mention the reaction as “solvothermal or hydrothermal” in figure1 it is mentioned as hydrothermal.

7.      Authors should explain the figure 3b, small particles present in the picture (impurity or broken particles)

8.      Authors explained, that FeOF can store more than 1 lithium or sodium (in line number 40) mechanism (equation) of lithium and sodium storage during the charge-discharge process is required.

9.      Figure 4e and Figure 5e need to be fitted with an Electrical equivalent circuit (also check the figure axis names).

10.   Line 214 please check and correct, sothiation?

11.   Line 219, the authors mention CEI, however, in the previous section it is mentioned SEI, please correct it and make it consistent. 

Reviewer 4 Report

This work synthesized a FeOF nanorod by hydrothermal methods and tested its performance in LIBs and SIBs. Overall, this paper looks like a simple experimental report without a thorough analysis and understanding of mechanism, process, and performance. Moreover, the manuscript is poorly written. I can find typos and grammar errors throughout this paper. In addition, Table 1 the author summarized to compare results with others is incorrect. For example, in reference 25, the battery was cycled at least 200 cycles. You listed it as 100 cycles. I cannot tolerate this kind of academic error in scientific publications. Therefore, I would not recommend it be published in its current form. I have some suggestions to help you improve the quality of the paper, such as investigating the kinetics difference between LIBs and SIBs by GITT analysis, and understanding the structure-to-performance relationship by regulating the FeOF morphology.